# Experimental Study on the Damage Characteristics and Acoustic Properties of Red Sandstone with Different Water Contents under Microwave Radiation

**DOI:** 10.3390/ma16030979

**Published:** 2023-01-20

**Authors:** Junjun Liu, Jing Xie, Bengao Yang, Fei Li, Huchao Deng, Zundong Yang, Mingzhong Gao

**Affiliations:** 1State Key Laboratory of Hydraulics and Mountain River Engineering, College of Water Resource & Hydropower, Sichuan University, Chengdu 610065, China; 2Guangdong Provincial Key Laboratory of Deep Earth Sciences and Geothermal Energy Exploitation and Utilization, College of Civil and Transportation Engineering, Shenzhen University, Shenzhen 518060, China

**Keywords:** microwave rock breaking, red sandstone, different water contents, damage characteristics, acoustic properties

## Abstract

Rock breaking is one of the most basic issues in deep underground engineering. Water plays an important role in the rock response under microwave radiation. Consequently, microwave radiation experiments using red sandstone with different water contents were conducted. The damage characteristics and ultrasonic properties of red sandstone after microwave radiation were primarily investigated, and the representative conclusions were drawn as follows: With the increase in water content, the time of complete formation of the rupture surface of the rock sample gradually decreased, and the decreasing range gradually increased. When the fracture surface is completely formed, the samples with a higher water content have more powdery rock cuttings and less surface roughness. The damage degree of the samples does not increase significantly with the increase in the water content when the sample is radiated at the same time. As the microwave radiation time is increased, the damage degree of the sample will increase significantly. Through the ultrasonic velocity test, it can be suggested that the sample exhibits obvious zonal damage characteristics under the action of a microwave. Generally speaking, it is a very effective means of improving the degree of microwave attenuation of the rock by increasing the water content of the rock mass.

## 1. Introduction

Deeper resource development and engineering construction are the inevitable trends of social and economic development [1,2,3,4,5]. However, the obvious increase in the strength of the deep rock body makes the development and utilization of the deep underground space face a great challenge, so there is an urgent need to innovate the existing rock-breaking technology to serve the deep resource development and engineering construction [6,7,8,9,10].

Microwave radiation technology is a highly promising technology for assisting rock breaking and releasing rock stress, which has great advantages for improving the construction efficiency and safety of deep engineering. Since the 1950s, microwave equipment has been developed, and after that, microwave technology has been widely used in the field of mining and metallurgy [11,12]. With the development of science and technology, microwave technology has become more and more mature, and microwave rock breaking has gradually become a hot spot of academic research in the field of deep underground engineering.

Rock mineral composition and microwave radiation conditions are the two main factors that can affect the effect of microwave rock breaking [13,14]. The former mainly includes mineral composition, mineral content, mineral distribution, microwave-sensitive mineral content and water content, which determines the ability of the rock to absorb microwaves and convert microwave energy into thermal energy [15]. The latter mainly includes the frequency and power of microwaves, which determine the distribution of the electromagnetic field energy domain. The ability of a rock to absorb microwaves can be roughly determined by the content of the various rock-forming minerals in the rock [16,17]. Rock-forming minerals can be divided into three categories according to their ability to absorb microwaves: strong microwave-absorbing minerals, medium microwave-absorbing minerals, and weak microwave-absorbing minerals [18]. After absorbing microwaves, the temperature of the rock increases, which can inevitably lead to thermal expansion and cracking. Li et al. [19] studied the change in mineral structure under microwave radiation and concluded that the mechanism of microwave rock breaking makes the rock produce irreversible thermal expansion and fracture. Ali and Bradshaw [20] and Osepchuk [21] found two causes for rock damage resulted from microwave radiation: (1) tensile stress caused by thermal expansion of minerals; and (2) shear stress caused by the intergranular slip of minerals after thermal expansion [22,23]. Hartlieb et al. [24] studied the mechanism of crack expansion under microwave radiation and found that the cracks inside the sample always developed along the radial direction.

After microwave radiation, cracks will be produced inside the rock, which is visualized as the reduction of rock strength, ultrasonic velocity, and wave energy [25,26]. Hartlieb et al. [27] found through experiments that the increase in microwave power and radiation time will increase the degree of damage to the rock, and the rock strength will be reduced. Kingman et al. [28] found that the strength of all types of rocks would be reduced under microwave radiation at different degrees, with the greatest reduction in igneous rocks. This verifies the possibility of microwave-assisted rock breaking from an experimental point of view. Lu et al. [29] independently developed an open microwave fracturing experimental device, used to conduct microwave-induced borehole fracturing experiments. Gao et al. [30] and Yang et al. [31] found that the rock strength and P-wave velocity were reduced at different degrees under microwave radiation. The results show that there is a certain degree of reduction in wave velocity and crack development in the rock mass around the borehole.

The permittivity is one of the most important parameters affecting the response characteristics of rocks under microwave radiation. However, the permittivity of water and the major rock-forming minerals differ by a factor of tens, so the water content in the rock will inevitably affect its response characteristics in the microwave field. This topic still needs more experiments and mechanism analysis. Microwave radiation experiments on red sandstone with different water contents were carried out to study the effect of water content on the fracture characteristics and ultrasonic properties of red sandstone under microwave radiation, with the view of providing some guidance for the application of microwave technology in the field of rock breaking.

## 2. Materials and Methods

Sandstone is a common rock in deep engineering. Due to the influence of geological environment and structure, its mechanical properties are quite different. This test uses relatively homogeneous red sandstone as material as far as possible to avoid the influence of the initial damage and heterogeneity of the sample on the test results. The red sandstone is mainly composed of feldspar (60%), quartz (30%), cuttings (4%), and interstitials (6%), which is sand-like structure and massive fabric. The cementation method is contact cementation. The natural water content is 0.858%.

Firstly, the red sandstone block was processed into a standard sample of *Φ*50 mm × *H*100 mm. Next, the basic physical parameters such as weight, diameter, height, and P-wave velocity of the sample are tested. The basic physical parameters of the samples are shown in Table 1.

In order to explore the effect of water content on the interaction between microwave and red sandstone, red sandstone samples with different water contents were prepared as follows.

Dry-state samples: the natural samples were placed in the oven at 105 °C for 24 h to obtain the dry-state sample.Samples soaked in water for 24 h: the natural samples were immersed in water for 24 h to obtain the target sample, which is mainly used for reference with the dry-state sample and saturated sample.Saturated samples: the natural samples were exposed to a vacuum for 24 h to obtain saturated sample.

Two experiments were carried out in this paper, including destructive experiment and microwave cyclic radiation experiment. Destructive experiment refers to the use of 1000 W microwave to radiate the sample to complete failure. Microwave cyclic radiation experiment refers to the use of 1000 W microwave to repeat radiation samples at fixed intervals. The P-wave velocity in the sample was measured after each microwave radiation. The microwave cycle radiation path is shown in Figure 1. The experiment process and main equipment are shown in Figure 2.

## 3. Results

### 3.1. Time for the Complete Penetration of the Fracture Surface of Samples with Different Water Contents

In order to explore the effect of the water content state of samples on microwave rocks alteration, four different water contents of red sandstone were prepared: a dry state, natural state, soaked in water for 24 h, and a saturated state. The samples with four different water-containing states were radiated with a 1000 W microwave until the fracture surface was completely penetrated, and the total time of microwave irradiation was recorded (see Table 2).

Figure 3 shows that the increase in the water content causes a gradual decrease in the time necessary for the complete damage of samples. In this paper, the complete formation time of the fracture surface is defined as the complete failure time.

The average water content of the natural samples was 0.858%, and their average complete destruction time under microwave radiation was 231.3 s. The average complete destruction time of the dried samples was 242 s, which was about 4.468% higher than the average complete destruction time of samples with the natural state. The average water content of the water-soaked 24 h samples was 1.312%, and their average complete destruction time under microwave radiation was reduced to 210.7 s, which was 8.91% lower than that of samples with the natural state. The average water content of the saturated samples was 1.704%, and the complete destruction time under microwave radiation plummeted to 181 s, which was 22.2% lower than the complete destruction time of the samples with the natural state. The results indicate that the water in the red sandstone accelerated the degradation of microwave degradation and can provide a new idea for microwave rock breaking.

### 3.2. Damage Characteristics of Red Sandstone with Different Water Contents

This section mainly focuses on exploring the variability of the fracture characteristics of red sandstone with different water content states under microwave radiation. The failure characteristics of red sandstone with different water contents under microwave radiation are shown in Figure 4.

With the effect of microwave radiation, the temperature of the sample is non-uniformly increased, which makes non-uniform stresses inside the rock, and cracks start to sprout when they exceed the crack initiation threshold. Under the continuous radiation of microwaves, the cracks continue to develop and penetrate at the same time. At the moment of complete formation of the fracture surface, it is accompanied by a strong explosion sound. It is worth noting that the explosion sound of the water-filled and water-soaked 24 h samples at the moment of complete rupture is much greater than that of the natural and dry samples.

As shown in Figure 4, a spherical melt with approximately the same volume is present on both of the end faces of the natural and dry samples after complete destruction. However, no melt was seen on the fracture surface of the samples soaked for 24 h and the saturated samples, but several smaller spherical melts were scattered around the samples. The rupture surfaces of the dry and natural samples have significant macroscopic cracks, and the cracks start from the spherical melt and extend to the boundary of the sample along the radial direction of the melt, while the rupture surfaces (the lower end surface) of the samples saturated with water and soaked in water for 24 h do not have macroscopic cracks. As the water content of the sample increases, the sound is also louder when the fracture surface is completely penetrated. In addition, a large amount of powder will be produced after the samples saturated with water and soaked in water for 24 h are completely destroyed, and the particle size of the powder generated by the saturated samples is smaller than that of the sample soaked in water for 24 h. These results indicate that the high-water content samples react more vigorously under microwave radiation, which leads to the conclusion that that water plays an important role in the process of rocks under microwave radiation.

It can also be seen from Figure 4 that the surface roughness of each sample is quite different. In order to qualitatively characterize the roughness of the fracture surface, a 3D scanner is used to obtain 3D images (See Figure 5) and elevation information of the local fracture surface. Four 8 mm × 6 mm rectangles are selected evenly along the circumferential direction on the fracture surface, the root mean square height of each region is calculated, and the average value is obtained. The root mean square height is used to characterize the surface roughness qualitatively. From high to low-water content, the roughness of the fracture surface is 601 μm, 959 μm, 992 μm, and 1150 μm, respectively. This shows that the roughness of the fracture surface decreases with the increase in the water content.

After reviewing the literature and analyzing the experimental phenomena, the driving factors of crack initiation and propagation in rocks under microwave radiation can be classified into the following four types: (1) After the mineral melts, the volume expands, which result in the extrusion force that is generated from the inside to the outside of the sample. (2) The free water and crystalline water inside the rock cannot escape quickly after vaporization, thus they can accumulate in the narrow pores and fissures to produce ultra-high air pressure. (3) The microwave radiation will cause a non-uniform temperature field inside the rock, which can result in thermal stress in the rock. (4) Each rock-forming mineral has a different thermal conductivity, thermal diffusivity, and thermal expansion coefficient, resulting in incompatible deformations between various minerals.

### 3.3. Micro-Level Damage of Rocks after Microwave Radiation

The difference in the macroscopic damage characteristics of rocks with different water contents is sure to closely relate to the damage at the microscopic level of the rock. Therefore, the natural water content samples before and after microwave action were observed under a crossed polarized light microscope, and the images are shown in Figure 6.

As shown in Figure 6, the red sandstone before microwave radiation has a lower degree of crack development on the whole, and there are a small number of transgranular and intergranular cracks. After microwave radiation, the overall crack development degree of red sandstone increases significantly, and the number of intergranular cracks and transgranular cracks experience a great increase. In addition, due to the local stress concentration of red sandstone caused by microwave radiation, the phenomenon of multi-crack intersection appears. It can be suggested from the observation results of red sandstone under the orthogonal polarizing microscope that microwave radiation can cause a large number of transgranular cracks and intergranular cracks in the red sandstone, and the intersection of multiple cracks will also occur locally. Compared with Figure 6, it was found that the crack content and distribution did not change significantly after the sample was soaked in water for 24 h. However, the number of cracks in the samples soaked in water for 24 h after microwave radiation increases significantly, especially the transgranular cracks and intergranular cracks. This is mainly because the movement of water molecules existing in pores and fractures is intensified after microwave radiation, which intensifies the development of cracks. This is the essence of the macroscopic damage phenomenon on rocks after microwave radiation.

### 3.4. Ultrasonic Properties of Rocks after Cyclic Microwave Radiation

Existing studies have shown that with the intensification of rock damage, the rock P-wave velocity gradually decreases. Therefore, previous research uses the change in the rock P-wave velocity to characterize the degree of damage on the rock. In order to explore the damage characteristics of samples with different water contents after microwave irradiation, three kinds of rocks with different water contents were used for microwave cyclic irradiation, and the P-wave velocity of the samples was tested after cooling.

It can be seen from Figure 7 that the P-wave velocity of the saturated samples increased from 3.301 km/s to 4.257 km/s, an increase of 29.0%. The ultrasonic velocity in solids, liquids, and air decreased sequentially. After saturating the samples, the pores are filled with water, so the P-wave velocity of the sample increased greatly. After drying, the wave speed of the samples increased from 3.303 km/s to 3.522 km/s, an increase of 6.6%. After the sample was dried at 105 °C for 24 h, the volume of the mineral matrix expanded to fill the primary cracks of the sample, but the damage threshold of the sample was not reached, which resulted in an increase in the P-wave velocity.

It can also be seen from Figure 7 that after the first microwave irradiation of the dried sample, the P-wave velocity decreased to 3.371 km/s, a decrease of 4.3%. In addition, the ultrasonic velocity of the natural sample dropped to 2.948 km/s, a drop of 10.4%, and the ultrasonic velocity of the saturated sample dropped to 2.887 km/s, a drop of 32.2%.

With the increase in the water content of the sample, the decrease in the P-wave velocity becomes larger after microwave irradiation, which indicates that the water in the sample stimulates the microwave rock breaking. Under microwave radiation, the free water and crystalline water in the rock will be rapidly vaporized but cannot be released in time. Therefore, it will lead to high pore pressure inside the rock, which promotes the development of damage.

The P-wave velocity of the samples in the dry and natural state does not show a decreasing trend in velocity after the second and third microwave irradiation. After the second and third microwave irradiation of the saturated sample, the P-wave velocity shows a weak downward trend, and the decrease range was only 1.6%. The rock matrix shrinks during cooling, which improves the ability of the sample to accommodate deformation. After microwave radiation, cracks connecting to the outside of the sample appeared inside the sample, causing water in the vaporized state inside the sample to escape to the outside of the sample through the cracks. When the sample is radiated with the same microwave parameters, the degree of damage basically does not increase.

When the microwave cyclic irradiation time was increased from 60 s to 120 s, the P-wave velocity of the dry, natural, and water-saturated samples all decreased significantly again; the decrease rates were 4.99%, 2.88%, and 7.37%, respectively. However, when the sample was continuously irradiated with microwaves for the same time, its P-wave velocity remained substantially unchanged. The experimental results also verified the above conclusions.

### 3.5. Damage Differences in the Different Regions of Red Sandstone under Microwave Radiation

The size and material of the reactor is affected by the location of the microwave source, which results in a non-uniform distribution of the microwave field intensity in the reactor, which will lead to the zonal damage characteristics of rocks under microwave radiation. This subsection mainly discusses the degree of damage of different regions in the sample under microwave irradiation.

The sample after microwave cyclic radiation is cut into five sections on average, and the ultrasonic velocity of each section of the sample is measured separately, and the wave velocity is shown in Figure 8. The P-wave velocity of the Z-9 sample in the natural state is 3.198 km/s. After five cycles of microwave radiation, the P-wave velocity is reduced to 2.89 km/s, a decrease of 9.6%. From top to bottom, the decreases in P-wave velocity were 7.4%, 14.0%, 16.9%, 12.4%, and 4.9% for each section of the specimen.

The overall trend of the experimental result shows that the P-wave velocity in the middle of the sample has a larger drop, and the P-wave velocity at both ends of the sample has a smaller drop. The ultrasonic velocity of each section of the sample after microwave cycle shows a huge difference, which indicates that the rock exhibits obvious regional damage characteristics under microwave irradiation. This characteristic is determined by the field strength distribution of the electric field that is excited by the microwave, the mineral distribution inside the sample, and the selective heating of the microwave. Since the red sandstone sample used in this test is relatively homogeneous, it can be considered that the non-uniformity of the sample has little effect on the test results. Therefore, it can be concluded that the regional damage characteristics of rocks under microwave radiation are mainly determined by the distribution of the microwave excitation electric field. In addition, this may be related to the uneven distribution of water in the sample, and the areas with the highest water content will have the greatest damage.

## 4. Discussion

As shown in Section 3, the damage and destruction characteristics of red sandstone samples with high and low-water contents under microwave radiation show significant differences. The differences mainly include the following: (1) The presence or absence of spherical molten material on the rupture surface. (2) The presence or absence of a large amount of powder production when the fracture surface is completely penetrated. (3) The time of the fracture surface penetration. (4) The acoustic emission phenomenon when the fracture surface is fully penetrated. The above differences will be discussed below.

It was found that a large amount of powder was produced when the fracture surface of the specimen with a high-water content was completely formed, accompanied by a large acoustic emission (see Figure 4). In contrast, the low-water content specimen did not produce powder when the fracture surface was completely penetrated, and it was accompanied by slight acoustic emission phenomenon at the same time. It is noteworthy that the melt on the fracture surface of the natural sample has a honeycomb structure and also has obvious jet traces, which is evidence of the presence of high-pressure water vapor inside the sample (see Figure 9).

Water will vaporize rapidly into gas under microwave radiation. However, due to the lack of fracture channels connected with the outside in the red sandstone sample, the gas cannot escape to the outside in time, resulting in high pore pressure in the sample. Owing to the presence of the fracture tip effect, the high pore pressure supports the expansion and penetration of cracks (see Figure 10), which accelerates the damage process of the red sandstone samples with a high-water content under microwave radiation.

It can be seen from Figure 6 that the bond cementation of red sandstone is contact cementation, and the cohesive force of this type is relatively weak. The spaces in the rock that are not filled by mineral particles, cuttings, or cement are called pores. The pore space in the rock is the place where oil, gas, and water are stored. Under microwave radiation, water is stored as water vapor in the pore space inside the sample, and the water is in a state of high-speed vibration. Vibrating water molecules in the pores will destroy the cementation structure of the matrix around the pores, so that the matrix around the pores loses cohesion. At the complete formation of the fracture surface of the sample, mineral particles that have lost their cohesive force are sputtered outward as the high-pressure gas is sharply released. This is the main reason why high-water content red sandstone samples produce large amounts of powder when the fracture surface is completely penetrated.

It can be seen from Figure 9 that the melt contains a certain amount of water vapor. High-water content red sandstone samples contain a lot of free water molecules, and the large number of vigorously moving water molecules can prevent the large-scale coalescence of the fluid melt to the extent that a large spherical melt cannot be formed. When the fracture surface is completely penetrated, the high pore pressure is sharply released, and the melt that has not coalesced on a large scale spills out. As the fluid melt has a certain cohesive force and surface tension, it spontaneously condenses into small particles of spherical melts again during the outward sputtering process (see Figure 11).

As shown in Figure 4, it can also be found that after the complete destruction of the high-water content red sandstone sample, the upper end of the sample is largely broken, but the lower end is relatively intact, and no macroscopic cracks are seen on the surface. The reason for this is that the microwave radiation time is short, the overall temperature of the sample is low, and the temperature gradient is small, which can result in a small, internal temperature stress of the sample. Secondly, since a large volume of spherical melts is not formed, it cannot produce a large squeezing pressure in the sample, which leads to the micro-cracks inside the sample not penetrating into macroscopic cracks.

## 5. Conclusions

By conducting microwave radiation tests on red sandstone with different water contents, the following conclusions were drawn.

(1) The time for complete penetration of the fracture surface of the red sandstone sample under microwave irradiation decreased with an increase in the water content, and the decreasing range gradually increased.

(2) Under microwave radiation, the damage characteristics of the samples with a high-water content and a low-water content showed great difference. The high-water content produces a large amount of powder when completely destroyed, and there is no spherical melt on the fracture surface. However, the low-water content red sandstone samples did not produce powder when complete destruction occurred and had spherical melts at both the upper and lower fracture surfaces. With the increase in the water content, the roughness of the rupture surface of the sample gradually decreases. In addition, as the water content increases, the energy of the acoustic emission time of the sample at the time of complete penetration of the rupture surface is greater.

(3) The ultrasonic properties of the red sandstone samples with different water contents after microwave radiation also showed large variability. The decrease in the P-wave velocity of the sample after microwave irradiation increases with the increase in the water content. In addition, when the fully cooled sample was irradiated repeatedly with the same microwave parameters (power and irradiation time), the decreasing trend in the P-wave velocity of the sample is not obvious.

(4) The rocks show obvious regional damage characteristics after microwave irradiation. The ultrasonic velocity of each part of the sample is negatively correlated with the electric field strength in the reaction chamber.

Wave attenuation is closely related to the fractures content and damage development in rocks, and the influence mechanism of microwave radiation on wave attenuation will be paid more attention in future research.

## Figures and Tables

**Figure 1 materials-16-00979-f001:**
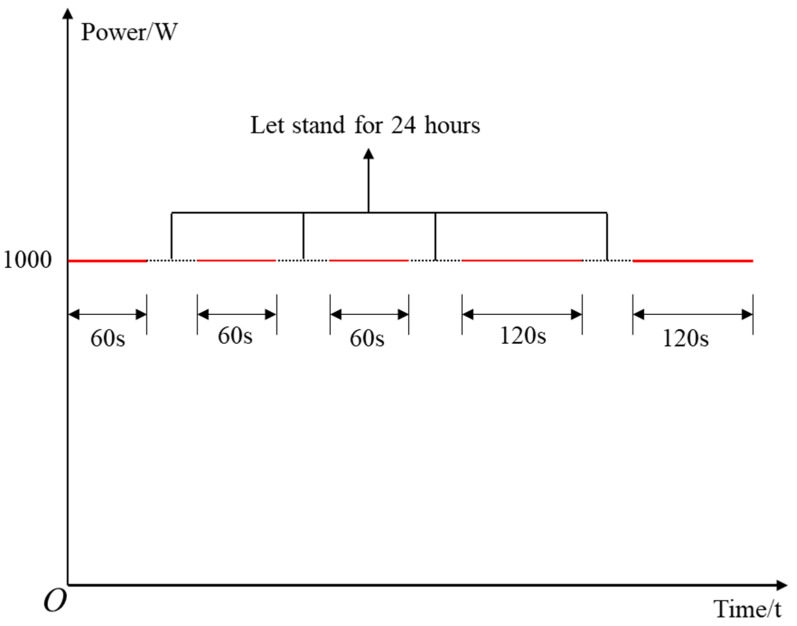
Microwave cycle radiation path.

**Figure 2 materials-16-00979-f002:**
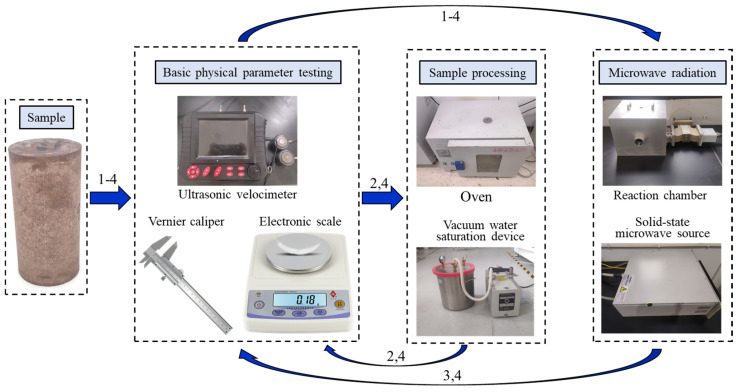
Experiment process and equipment.

**Figure 3 materials-16-00979-f003:**
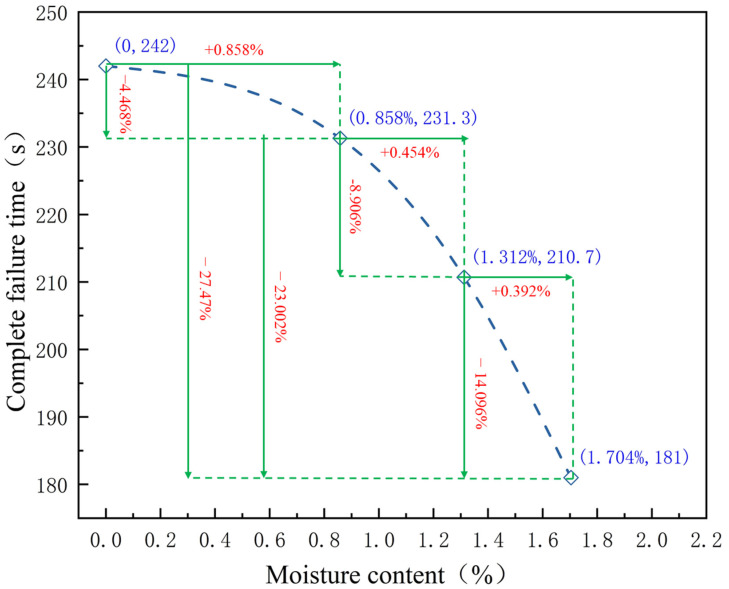
Relationship between water content of samples and complete failure time.

**Figure 4 materials-16-00979-f004:**
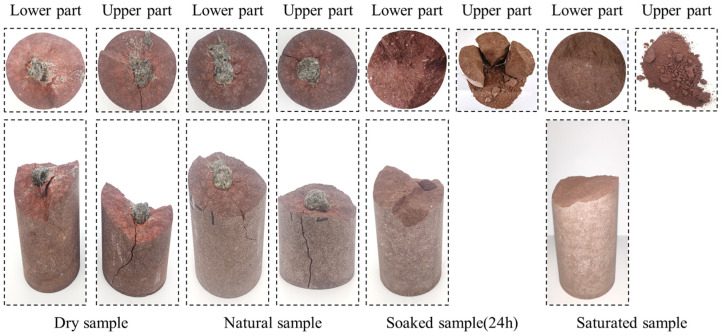
Fracture characteristics of red sandstone with different water contents.

**Figure 5 materials-16-00979-f005:**
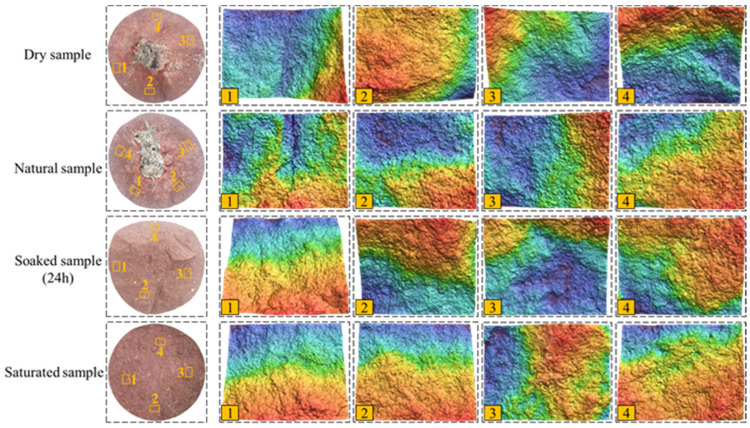
Three-dimensional scanning images of local area of fracture surface.

**Figure 6 materials-16-00979-f006:**
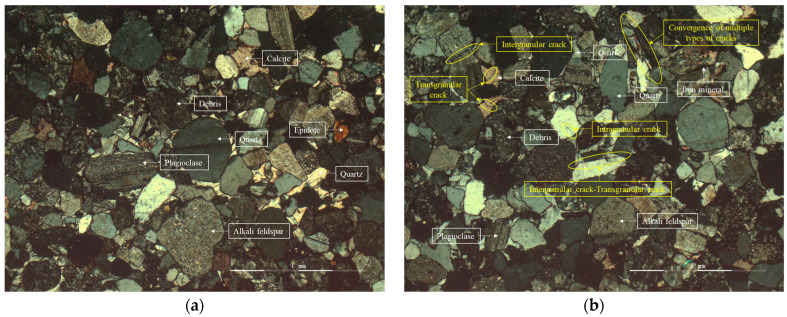
Orthogonal polarization image of sandstone samples: (**a**) natural water content sample before microwave radiation, (**b**) natural water content sample after microwave radiation, (**c**) sample soaked in water for 24 h before microwave radiation, and (**d**) sample soaked in water for 24 h after microwave radiation.

**Figure 7 materials-16-00979-f007:**
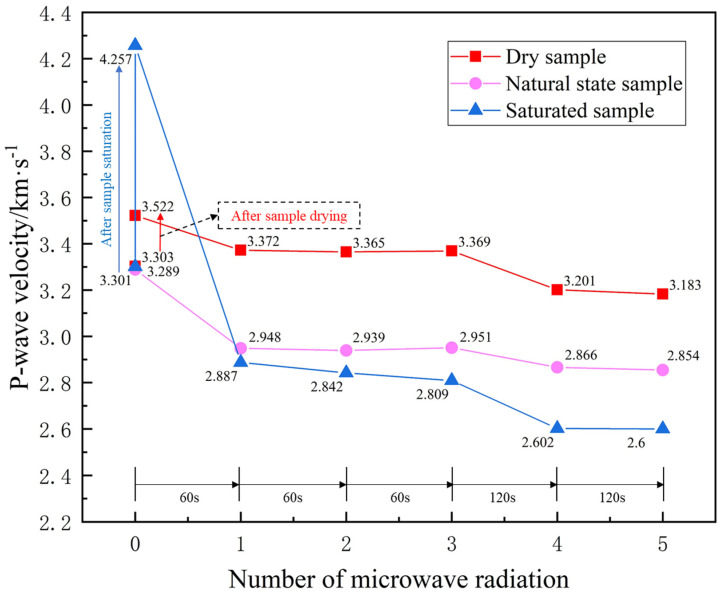
P-wave velocity of sample after microwave cyclic irradiation.

**Figure 8 materials-16-00979-f008:**
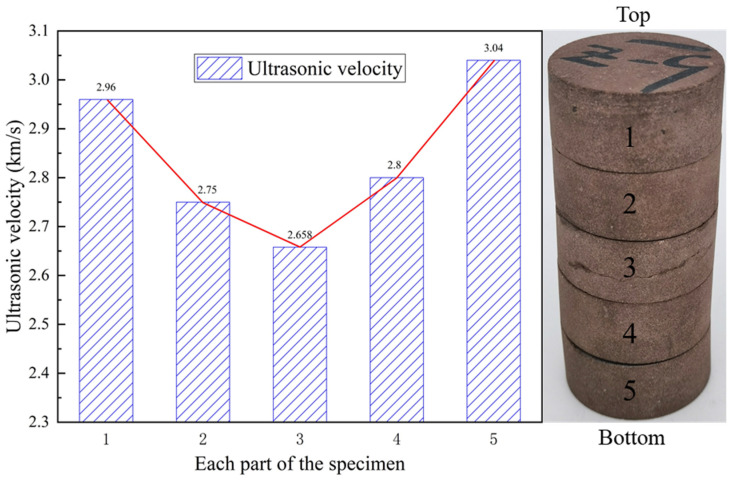
P-wave velocity of each section of the sample after cyclic microwave irradiation.

**Figure 9 materials-16-00979-f009:**
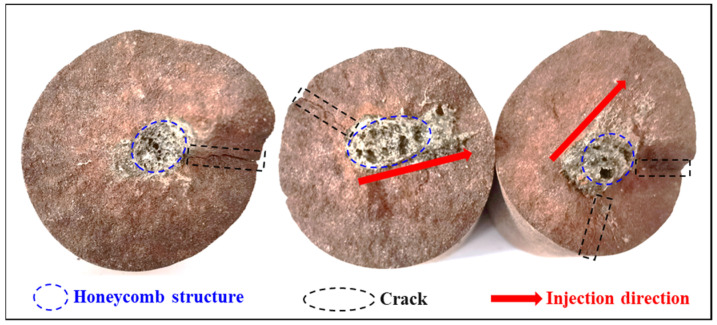
Honeycomb structure and jet traces on the fracture surface of the natural specimen.

**Figure 10 materials-16-00979-f010:**
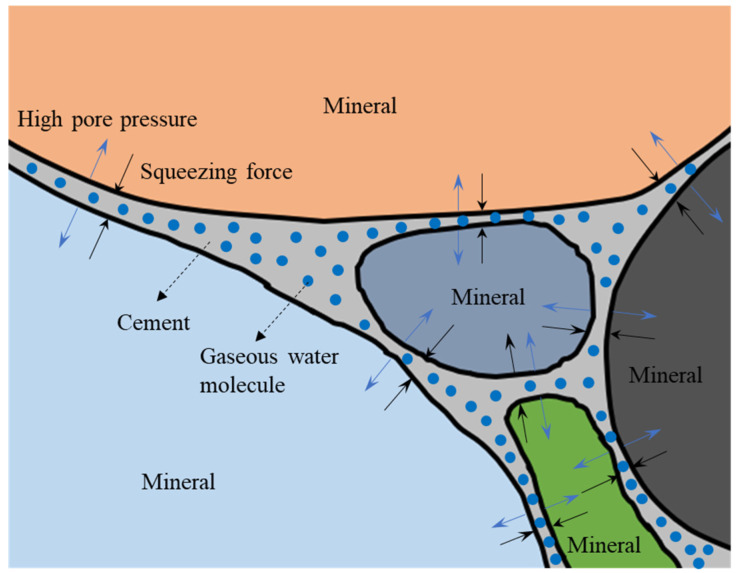
Damage mechanism of red sandstone under microwave radiation.

**Figure 11 materials-16-00979-f011:**
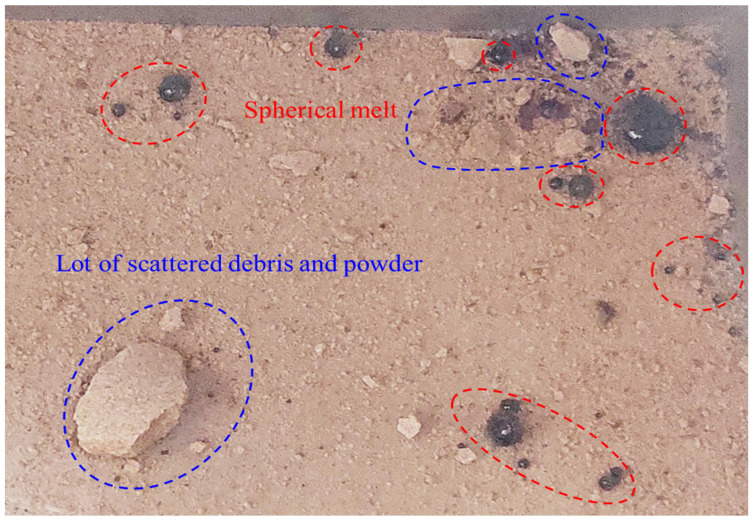
Scattered red sandstone fragments and spherical glassy melts.

**Table 1 materials-16-00979-t001:** Basic physical parameters of samples and their experimental use.

Water State	Sample Number	Height /mm	Diameter/ mm	Mass /g	Density /g·cm^−1^	P-Wave Velocity/ km·s^−1^	Water Content
Natural state	Z-0	100.21	49.81	490.52	2.512	3.218	0.863%
	Z-1	100.04	49.54	479.68	2.488	3.205	0.798%
	Z-2	100.06	49.53	479.28	2.486	3.219	0.913%
Dry state	21	100.12	49.32	479.08	2.505	3.226	-
	22	100.20	49.56	480.63	2.486	3.212	-
Saturated state	23	100.28	49.84	491.42	2.512	3.061	1.710%
	24	100.15	49.56	479.08	2.479	3.212	1.687%
	25	100.08	49.83	490.39	2.513	3.145	1.750%
	26	99.97	49.88	493.41	2.526	3.318	1.781%
	27	100.36	50.12	497.09	2.511	3.192	1.592%
Soaked in water for 24 h	28	100.11	49.55	482.93	2.502	3.165	1.324%
	29	100.12	49.55	484.18	2.508	3.185	1.295%
	30	99.98	50.14	496.09	2.513	3.290	1.317%
Dry state (Circular radiation)	Z-3	99.72	49.21	478.92	2.525	3.247	-
	Z-4	100.56	49.26	482.8	2.519	3.417	-
	Z-5	100.28	49.43	483.77	2.514	3.394	-
Natural state (Circular radiation)	Z-6	100.26	50.16	497.52	2.511	3.289	0.847%
	Z-7	100.16	49.29	480.23	2.513	3.276	0.720%
	Z-8	100.10	49.41	482.54	2.514	3.213	0.889%
	Z-9	100.16	49.58	486.05	2.513	3.198	0.904%
	Z-10	100.41	50.14	497.78	2.511	3.219	0.865%
Saturated state (Circular radiation)	M-11	100.35	49.07	478.92	2.524	3.262	1.712%
	M-12	100.25	49.33	481.05	2.511	3.247	1.705%
	M-13	100.54	49.26	481.48	2.513	3.320	1.713%

**Table 2 materials-16-00979-t002:** Water content and failure time of the sample.

Water State	Average Water Content	Sample Number	Complete Destruction of Time	Average Destruction Time /s
Natural state	0.858%	Z-0	238	231.3
		Z-1	230	
		Z-2	226	
Dry state	0	21	240	242.0
		22	244	
Soaked in water for 24 h	1.312%	28	220	216.7
		29	218	
		30	212	
Saturated state	1.704%	23	160	181
		25	185	
		27	198	

## Data Availability

Not applicable.

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
