# Peer review of "Experimental Study on the Damage Characteristics and Acoustic Properties of Red Sandstone with Different Water Contents under Microwave Radiation"

_materials, 2023, doi:10.3390/ma16030979_

Round 1

Reviewer 1 Report

1. Line 160, “Figure 4 that the roughness of the rupture surface shows a significant decrease with the increase of water content of the sample”. How to calculate the roughness of the rupture surface?

2. Fig. 5b, what is the meaning of “How degree of crack development” ? In this figure, the differences in crack development before and after microwave radiation are demonstrated. However, the influences of water content on the crack development is not provided.

3. It is difficult to under that the P-wave velocity of saturated specimen is lower than that of dried specimen.

4. In the Discussion, the acoustic emission phenomenon is discussed. However, the results of acoustic emission are not presented in this manuscript.

5. Please check carefully for the whole manuscript. The manuscript is encouraged to make a correction by an English editor.

Reviewer 2 Report

The paper studies the effect of microwave radiation on rock damage and its acoustic properties. The paper investigated the effect by changing the moisture degree, starting by natural case, dry, saturated and sample 24 h soaked in water. Overall, the outcomes of the paper are good, as well as the interpretation. However, many areas needs improvement, that is why I need a second reading of the paper after that authors implement my suggestions and corrections, and also their arguments if they disagree with some of them. I would like that the below point to be addressed.

-     -    It is still unclear to me what the authors mean by penetration of the fracture, do you mean when fractures connect to rock pores?

-  -  The authors show the variation of ultrasonic velocity according to the moisture degree (dry, natural, etc.). In my opinion the error in velocity measurements should be discussed, which is necessarily related to accuracy in time picking.

-  -  Why the authors have not investigated the effect of microwave radiation on shear velocity? It should be very interesting to see the radiation effect on Vp/Vs as well.

-   -   It should be good and more interesting, if the authors investigated the effect of microwave radiation on wave attenuation, which is physically closer to rock damages than velocity. At this stage I suggest that the author add the below sentences or similar to it, at the end of the conclusion:

“Wave attenuation is closely related to fractures content and damage development in rocks than velocity, because of scattering and fluid-related mechanisms (e.g. Matsushima et al., 2017; Bouchaala et al., 2022). Investigation of the effect of microwave radiation should be considered in future similar studies”

·         Matsushima, Jun, Mohammed Y. Ali, and Fateh Bouchaala. "A novel method for separating intrinsic and scattering attenuation for zero-offset vertical seismic profiling data." Geophysical Journal International 211, no. 3 (2017): 1655-1668.

·         Bouchaala, Fateh, Mohammed Y. Ali, Jun Matsushima, Youcef Bouzidi, Mohammed S. Jouini, Eric M. Takougang, and Aala A. Mohamed. "Estimation of Seismic Wave Attenuation from 3D Seismic Data: A Case Study of OBC Data Acquired in an Offshore Oilfield." Energies 15, no. 2 (2022): 534.

-    -     English editing is needed, although that I did lot of editing on the attached file.

- Many comments and corrections are added in the attached document. Please, read them carefully and do the adequate corrections.

Author Response

Please see the attchment.

Round 2

Reviewer 1 Report

All comments have been responsed. No further suggestions.

Reviewer 2 Report

The authors made lot of changes and answered all the questions which significantly improved the manuscript. I am satisfied with their review, and thank you for the video.